# Recovery of Ionospheric Signals Using Fully Convolutional DenseNet and Its Challenges

**DOI:** 10.3390/s21196482

**Published:** 2021-09-28

**Authors:** Merlin M. Mendoza, Yu-Chi Chang, Alexei V. Dmitriev, Chia-Hsien Lin, Lung-Chih Tsai, Yung-Hui Li, Mon-Chai Hsieh, Hao-Wei Hsu, Guan-Han Huang, Yu-Ciang Lin, Enkhtuya Tsogtbaatar

**Affiliations:** 1Department of Space Science and Engineering, National Central University, Taoyuan City 320317, Taiwan; mmmendoza@g.ncu.edu.tw (M.M.M.); jason0010125@g.ncu.edu.tw (Y.-C.C.); chlin@jupiter.ss.ncu.edu.tw (C.-H.L.); davidhsieh@g.ncu.edu.tw (M.-C.H.); willy108623016@g.ncu.edu.tw (H.-W.H.); enter468@g.ncu.edu.tw (G.-H.H.); alterjohnnylife@g.ncu.edu.tw (Y.-C.L.); 2Skobeltsyn Institute of Nuclear Physics, Lomonosov Moscow State University, 119899 Moscow, Russia; 3Center for Space and Remote Sensing Research, National Central University, Taoyuan City 320317, Taiwan; lctsai@csrsr.ncu.edu.tw; 4AI Research Center, Hon Hai Research Institute, Taipei 114699, Taiwan; yunghui.li@foxconn.com; 5Department of Computer Science and Information Engineering, National Central University, Taoyuan City 320317, Taiwan; enkhtuya@g.ncu.edu.tw

**Keywords:** ionospheric sounding, space weather, artificial intelligence, fully convolutional DenseNet

## Abstract

The technique of active ionospheric sounding by ionosondes requires sophisticated methods for the recovery of experimental data on ionograms. In this work, we applied an advanced algorithm of deep learning for the identification and classification of signals from different ionospheric layers. We collected a dataset of 6131 manually labeled ionograms acquired from low-latitude ionosondes in Taiwan. In the ionograms, we distinguished 11 different classes of the signals according to their ionospheric layers. We developed an artificial neural network, FC-DenseNet24, based on the FC-DenseNet convolutional neural network. We also developed a double-filtering algorithm to reduce incorrectly classified signals. That made it possible to successfully recover the sporadic E layer and the F2 layer from highly noise-contaminated ionograms whose mean signal-to-noise ratio was low, SNR = 1.43. The Intersection over Union (IoU) of the recovery of these two signal classes was greater than 0.6, which was higher than the previous models reported. We also identified three factors that can lower the recovery accuracy: (1) smaller statistics of samples; (2) mixing and overlapping of different signals; (3) the compact shape of signals.

## 1. Introduction

The task of monitoring the ionosphere above the Taiwan–Korea region is important for the control and prediction of the conditions for the propagation and distortions of signals from the Global Navigation Satellite Signal (GNSS) constellation. The low-latitude ionosphere is highly variable due to the daily, seasonal, and solar cycle variations, as well as due to the impacts of the external and internal drivers. These external drivers are related to space weather effects such as solar flares [1], geomagnetic storms [2], and particle precipitations from the radiation belts [3]. The internal sources are related to tidal effects [4], earthquakes [5], and tsunamis [6]. Operative identification of the ionospheric disturbances of different origins is required for the development of nowcast and forecast models of the regional ionosphere. The wide variety of disturbances and the complex behaviors of the ionosphere require the development of advanced methods of ionospheric observation and diagnostics.

Ionospheric sounding is conducted by using the standard techniques of ground-based ionosondes and a network of Global Positioning System (GPS) receivers, which provide high temporal resolution. Ionosondes can measure the vertical profiles of electron density from the ground up to the maximum of the F2 layer [7], while the GPS receivers can provide the Slant Total Electron Content (STEC) by estimating the signal delay from the satellites to ground-based receivers [8]. Unfortunately, the coverage of these standard instruments is not sufficient in the Taiwan–Korea region mainly because the instruments can not be installed in the sea. In order to enhance spatial coverage, a novel technique known as the Vertical Incidence Pulsed Ionospheric Radar (VIPIR) was implemented in Korea and Taiwan. VIPIR is a digital ionosonde consisting of several antennas [9].

Active sounding allows the determination of important local ionospheric parameters for regional modeling of the ionosphere and for calculation of ionospheric indices. The oblique sounding with sweeping frequency will allow the measurement of the vertical ionization profile above the sea surface. Oblique sounding is one the most effective techniques for the observation of Traveling Ionospheric Disturbances (TIDs) and the determination of their properties and origin. The technique can also verify the assumptions of a uniform and quasi-static reflecting ionospheric boundary. The ionospheric scintillations could also be estimated from fast fluctuations of the received signal.

Ionogram recovery from the aforementioned vertical and oblique sounding technique is a very complex problem. The raw ionogram contains many parasitic signals from numerous ground-based radio transmitters. Thus, a special data processing technique is required in order to recover the useful ionospheric signal from the contaminations and the noise. Currently, ionogram recovery methods are based on fuzzy segmentation and connectedness techniques [10]. These methods however cannot completely remove the noise from the ionograms, leading to further manual data treatments. The VIPIR technique produces thousands of ionograms per day. The amount of raw data from two VIPIRs and two dynasondes is estimated to be about 20 Terabytes (TB) per year. It is almost impossible to operate on these data manually.

In recent years, deep learning techniques have been implemented to aid ionospheric research. Predictive models based on deep learning methods have been developed in order to forecast GPS-TEC at midlatitudes in Turkey [11]. In another similar study, Convolutional Neural Networks (CNNs) were also used to forecast TEC values globally [12]. Deep learning has also played an important role in the recent Coronavirus Disease 2019 (COVID-19) pandemic. In particular, the attention-based VGG-16 model has been used as an image classification tool for COVID-19 diagnosis [13]. Recently, CNNs have been used to automatically recover the signals of the ionospheric F-layer from experimental ionogram data in Peru [14]. The results in that paper showed that CNN models significantly removed the noise from the recovered ionograms, and thus, the quality of information was improved.

In this study, we approached the problem of ionogram recovery by implementing a CNN model known as FC-DenseNet [15]. We show numerically that ionogram data obtained from Taiwan ionosondes have far more noise and contaminations as compared to Peru data. In addition, this image segmentation technique, which separates the wanted signals such as the ionogram signals from the unwanted signals such as the background and the noise, can also provide a classification between signals from various ionospheric layers as different classes. We demonstrate that the model has different prediction capabilities for different classes because of various reasons, which are identified and discussed.

The goal of this work was to present the application of an advanced CNN model for improvement of the ionogram recovery techniques, as well as to present several challenges upon applying the model for the complex ionogram data. Section 2 presents the experimental data together with their main characteristics and problem features. The CNN model is introduced in Section 3. Section 4 describes the results of the model application and discusses the problems for future work. Lastly, Section 5 is the conclusions.

## 2. Experimental Data from Ionosondes

### 2.1. Ionograms

An ionogram is a type of data obtained from an ionosonde. It is used to examine the vertical profile of the ionosphere as a function of frequency at a given time. The ionograms used to train the model were from ionosondes located at Hualien, Taiwan ($23.99^{\circ }$ N, $121.61^{\circ }$ E), and had a time resolution of 12 min. An example of an ionogram obtained by using the VIPIR technique is shown in Figure 1. It is represented in the form of a gif image (*f*, *h*, *I*), where *f* is the frequency, *h* is the virtual height, *I* is the intensity of the received signal represented in a color scale from 0 to 255, and the dimension of the ionogram images is 1600 by 800 by 256, respectively. This sample ionogram was obtained by using the VIPIR technique as mentioned in the Introduction. In this method, the antennas are situated in a specific order to determine the inclination and the azimuth of the incident radio waves. VIPIR can also transmit a radio signal with a sweeping frequency range from 1–20 MHz. This radio signal reflects from the bottom-side ionosphere at different heights depending on the frequency, which makes it possible to measure a vertical profile of the ionospheric electron density.

In the ionogram shown in Figure 1, we can distinguish a number of features: background $\left(S_{bg}\right)$, contamination $\left(S_{n}\right)$, calibration signal, and ionospheric signals (S). The background is a field of diffusive signals of low intensity. It can be defined as the instrumental noise that varies with time and frequency. The contamination manifests as intense vertical lines. This signal is produced by transmitters of constant-frequency radio signals. The calibration signal gives a thick horizontal line at the virtual height of 50 km. The ionospheric signals appear as curves and spots of intermediate intensity. Those signals are reflected from the ionosphere back to the ionosonde. In Figure 1, one can see the ordinary and extraordinary signals from the E, F1, F2, and F3 ionospheric layers. The extraordinary signals have a higher frequency at the given virtual height. The ionospheric signals are the useful signals, which we have to analyze. In addition, a secondary echo from ionospheric layers can be seen at heights above 500 km. Secondary echoes occur when the ionosonde signal reflected from the ionosphere travels back to the Earth’s surface then propagates back to the ionosphere, which reflects it back to the Earth. This double reflection produces the secondary echoes, whose virtual height is two-times higher than the actual height of the ionospheric layers. The main purpose of the ionogram recovery technique is to separate the ionospheric signals from the background, contamination, and secondary echoes.

In this work, the signals from ionospheric layers were labeled in 6131 ionograms from 2013 and 2014. The labeling procedure was performed by manually identifying the ionospheric layers for the the ordinary and extraordinary modes of the E, E_s_, F1, F2, and F3 layers on the ionogram. As a result, we obtained 11 different classes of ionospheric signals, as shown in Table 1.

The determination of the critical frequencies for the ionospheric layers is important to determine the amount of ionization in the ionosphere. In Figure 2, the distribution of the maximum and minimum frequencies of the labeled F2 layer of the ionograms is shown. The coverage of the critical frequencies was nonuniform, wherein several deep and wide gaps could be observed. This was owed to the presence of strong quasi-static man-made contamination signals from ground-based transmitters. We can also observe that the distribution of the maximum and minimum frequencies overlapped somewhere between 5 MHz and 8.5 MHz. This is because of the wide dynamics of the F2 layer. In order to address this issue, the ionogram recovery model should (1) operate inn a wide frequency range and (2) have a smart interpolation of the ionospheric layers.

### 2.2. Signal-to-Noise Ratio

To evaluate the level of the contamination, we define the Signal-to-Noise Ratio (SNR) of an ionogram based on the statistical distribution (i.e., histogram) of the signal amplitudes (pixel color *I*), that is the total occurrence number N(I) vs. *I*. The idea is to define a characteristic amplitude for the background $S_{\mathrm{bg}}$, contamination $S_{n}$, and ionospheric signal *S*, respectively, based on the statistical distribution and use the characteristic amplitudes to define the SNR.

Figure 3 (upper panel) shows the histogram of the pixel intensities *I* of an entire ionogram. The ionogram shows that $S_{\mathrm{bg}}$ was the weakest signal that occupies the largest portion of the image. The second most populated signal was $S_{n}$, of which the pixel intensity was comparable to that of *S*. Based on these observations, we expected that the major peak ($I_{\mathrm{mp}}$) in the histogram would be primarily contributed by $S_{\mathrm{bg}}$ and $S_{n}$. Since $S_{\mathrm{bg}}$ correspond to weaker signals, they should mostly be located at the left side of the major peak. Therefore, we can define the characteristic pixel intensity of $S_{\mathrm{bg}}$ ($CS_{\mathrm{bg}}$) as the median amplitude of all pixels with a pixel intensity smaller than $I_{\mathrm{mp}}$:$$CS_{\mathrm{bg}}\equiv \mathrm{Median}\left(I<I_{\mathrm{mp}}\right)$$

Since $S_{n}$ is composed of the stronger signals, we can define its characteristic pixel intensity ($CS_{n}$) as the median amplitude of all pixels with intensities higher than $CS_{\mathrm{bg}}$:$$CS_{n}\equiv \mathrm{Median}\left(I>CS_{\mathrm{bg}}\right)$$

To determine the characteristic pixel intensity for the useful signal, *S*, we considered the statistical distribution of the pixel intensity inside the labeled signals ($I_{s}$) and used their median value as the representative pixel intensity (CS):$$CS\equiv \mathrm{Median}(I_{s})$$

An example is shown in the lower panel of Figure 3. It can be seen that the distribution contained several peaks. The median of distribution (CS=53) fell into the major peak located between I=50 and I=60. Hence, the median value was robust for the statistical representation of the signal intensity.

Finally, the signal-to-noise ratio is defined as the ratio between the characteristic signal (CS) and characteristic man-made noise ($CS_{n}$):$$\mathrm{SNR}\equiv \frac{CS}{CS_{n}}$$

The distributions of the SNR are presented in Figure 4. The histogram for Taiwan data of 6131 ionograms is shown by orange. For comparison, we show the histogram for Peru data by blue. The Peru dataset consisted of 816 manually labeled ionograms acquired from Jicamarca ionosonde (see the details in [14]). It can be seen clearly that the SNRs of Peru data, with a mean SNR = 2.09, were predominantly higher than those of Taiwan data, with a mean SNR = 1.43. In other words, almost all Peru ionograms were cleaner than the Taiwan ionograms.

Figure 5 shows the comparison of the maximum and minimum SNR cases for Peru and Taiwan. We can see that the ionospheric signals in the noisiest Peru ionogram case, which had an SNR = 1.56, were still clearly distinguishable from the noise and background. However, it was difficult to determine the ionospheric signals for the noisiest Taiwan ionogram case, which had an SNR = 0.8.

### 2.3. Shape Parameter

An ionospheric signal can be classified into two types according to its shape. Figure 6 shows a diagram of the (a) compact signal and the (b) elongated signal. The area of the signal is shown in red, and the labeling area is shown in purple. The labeling area is located along the outer boundary of the signal; therefore, in most cases, the area enclosed by the labeling is larger than the signal. If the area of the labeling becomes comparable to the area of the signal, it may cause a significant decrease in the prediction performance. Conversely, if the area of the labeling is very small such that the signal is almost perfectly enclosed by the labeling, then the prediction accuracy is expected to be high. It is important to note that the accuracy of the model can never attain 100% since the process of manually labeling the signals introduces room for error. In order to evaluate how the shape of an ionogram signal affects the performance of the model, we define a shape parameter. The detailed derivations for the shape parameter is presented in Appendix A.

The shape parameter, which we refer to as the circumference-over-area C/A, distinguishes between the compact and the elongated signals. Compact signals will have a low C/A, while elongated signals will have a high C/A.

The performance of an image segmentation model is often measured in terms of the Intersection over Union (IoU) (Equation (Equation 1)).
(1)$$\begin{array}{c} \mathrm{IoU}=\frac{\mathrm{prediction}\cap \mathrm{ground truth}}{\mathrm{prediction}\cup \mathrm{ground truth}} \end{array}$$
(2)$${\mathrm{IoU}}_{\mathrm{perfect}}=\frac{A_{\mathrm{signal}}}{A_{\mathrm{signal}}+A_{\mathrm{label}}}$$

In this study, the area occupied by the ground truth was equal to the sum of the labeling area $A_{\mathrm{label}}$ and the signal area $A_{\mathrm{signal}}$, i.e., $A_{\mathrm{label}}+A_{\mathrm{signal}}$. In the case of a perfect prediction, that is the model prediction is exactly equal to $A_{\mathrm{signal}}$, the IoU would become Equation (Equation 2). The equation shows that the IoU for a perfect prediction would be smaller than 1 since $A_{\mathrm{label}}+A_{\mathrm{signal}}>A_{\mathrm{signal}}$. In Appendix A, we show that $A_{\mathrm{label}}+A_{\mathrm{signal}}$ is closer to $A_{\mathrm{signal}}$ for features with higher C/A. Therefore, the higher C/A is, the higher IoU for a perfect prediction. The elongated signals (high C/A) were expected to have a higher IoU compared to the compact signals.

## 3. Materials and Methods

### 3.1. Convolutional Neural Networks

CNNs are a special class of Artificial Neural Networks (ANNs) that are commonly used in computer vision applications. CNNs comprise multilayer networks that are specifically designed to process two-dimensional data such as images or videos [16]. The first application of CNNs were from the 1990s where the technique was applied to perform the task of hand-written character recognition [17]. The main advantage of CNNs over other kinds of ANNs is due to its capability of weight sharing, which significantly reduces the number of parameters needed for training a model. Hence, CNNs can be trained more smoothly and are less prone to overfitting [18]. Due to their remarkable performance, CNNs have been widely used in many areas of image recognition [19,20,21].

In terms of applying image segmentation techniques on ionograms, Reference [14] already used the Peru dataset to implement a model based on an encoder-decoder, also known as an auto-encoder, CNN architecture. The auto-encoder architecture used in their study had 4385 parameters, of which 4385 were trainable and 0 non-trainable. This consisted of 6 convolutional layers with a ReLU activation, 3 max pooling layers with zero padding, and 3 upsamplings. They used binary cross-entropy as their loss function and adadelta as an optimizer.

### 3.2. Fully Convolutional DenseNet

Fully Convolutional DenseNet (FC-DenseNet) is a Fully Convolutional Network (FCN) model evolved from a CNN model called DenseNet. Our specific model, FC-DenseNet24, was created based on FC-DenseNet103 [15]. The basic building blocks of FC-DenseNet103 are the Dense Block (DB), Transition Down (TD), and Transition Up (TU). The composition layers of DB, TD, and TU are shown in Table 2.

Since one of the purposes of our work was to make the results open to everybody, we provide our model and dataset on Kaggle (The data and models are provided in the Data Availability Statement section at the end), a web-based data and computing environment. The original version of the FC-DenseNet-103 model (see Figure 7, right panel) was too heavy to be run on Kaggle because of computing and memory limitations. Note that the resolution of Taiwan ionograms was 1600 × 800 pixels, and the number of samples was more than 6000. These data require much space; hence, we simplified the original model step-by-step in order to fit the requirements of Kaggle. As a result, we achieved a configuration of the FC-DenseNet-24 model (see Figure 7, left panel), which is friendly for Kaggle such that everybody can run the model, verify our results, and check for other datasets. For the adjustments, we reduced the number of blocks by two and the number of layers several times such that the number of training parameters was reduced from 9.4 million to 690,886.

As CNNs become increasingly deep, they can transfer more information. However, as information about the input or gradient passes through many layers, it may vanish [22] or explode [23]. In order to avoid that, FC-DenseNet24 has two mechanisms, which are the randomly dropping layers and the shortcuts, as depicted in Figure 8.

For the deeper CNNs, randomly dropping layers during training can allow better information and gradient flow. FC-DenseNet24 has a dropout with p=0.2 in the DB. FC-DenseNet24 distills this insight into a simple connectivity pattern: to ensure maximum information flow between layers in the network, we connected all layers (with matching feature map sizes) directly to each other. To preserve the feed- forward nature, each layer obtains additional inputs from all preceding layers and passes on its own feature maps to all subsequent layers. By this mechanism, FC- DenseNet24 requires fewer parameters than traditional convolutional neural networks, as there is no need to relearn redundant feature maps. The total parameters for our FC-DenseNet24 were 690,886, with 685,176 trainable parameters and 5710 nontrainable parameters.

### 3.3. Filtering Algorithm

The two most common incorrect predictions from our model were found to be incorrect classifications and secondary echoes mistaken as signals. To reduce such errors, we developed a double-filtering scheme described below:

#### 3.3.1. Filter for Incorrect Classification

Incorrect classification is when the class of a signal is incorrectly identified by the model. This can be reduced by applying the following boundary filter:

Based on the physical properties and statistical distributions of the signal positions in the ionograms, we first determined the upper and lower boundaries in the frequency and virtual height for each signal class. Any predicted signal pixels that were located outside of the boundaries were then considered as incorrect classification and removed;

#### 3.3.2. Filter for Secondary Ionospheric Echoes

We created the following algorithm to filter out the secondary ionospheric echoes that were incorrectly identified as signals:

The filter scans through each frequency bin. If there are *k* predicted pixels at a frequency bin $f_{i}$, the mean height of these *k* pixels is computed and rounded to the nearest height pixel. If these *k* pixels are all part of the same signal, their height pixels would be continuous, and the mean height pixel would be equal to one of them. In this case, the filter would consider there to be no echo to remove at this frequency bin. If the mean height pixel does not belong to the set of the *k* height pixels, this indicates that there is a gap among these pixels. The filter would consider any pixels located higher than the mean height pixel as the secondary echoes and remove them.

## 4. Results

In order to train the model, the dataset was divided into training, testing, and validation sets. Table 3 shows the percentages of each layer for the datasets.

Because of the low percentages of the eso and esx classes, the three classes assigned to the Es layer (eso, esx, es) were combined into one class “esa”, as presented in Table 3 and Table 4.

The accuracy level of the model prediction as evaluated by the IoU, defined in Equation (Equation 1). The IoU of our model prediction for different signal classes before and after the double filtering procedure is shown in Table 4.

In Table 4, we see that the model was capable of identifying the es and fbo signals with an accuracy of IoU >0.5. By applying the double-filtering algorithm, the accuracy was further increased to IoU >0.6. The recovery of the eso, esx, fao, fax, and fbx signals were between 0.2< IoU <0.5. The eo, ex, fco, and fcx classes had an IoU of zero.

The Peru group [14] applied an auto-encoder to recover the signals from the ionospheric F-layer without distinguishing different classes (such as F1, F2, and F3) and yield an IoU =0.57. In contrast, our model can predict the E and Es layers in addition to the F1, F2, and F3 layers. The former ones have a very different geometry and location from those for the F layer. Because of that, a simple binary classification cannot be applied for the comparison of the two models. Instead, we calculated the prediction of our model for the whole F layer, which was the combination of the following classes: F = fao + fax + fbo + fbx + fco + fcx. As a result, we obtained an IoU =0.7. Apparently, this accuracy was much higher than that for the Peru auto-encoder model. Our results show that despite operating on the ionograms with a much lower SNR and needing to distinguish different classes of signals, the FC-DenseNet24 model was capable of producing a much higher level of accuracy for the F layer than that of Peru’s auto-encoder.

In Figure 9, two examples of the double-filtering results are shown for the incorrect classification filter and in Figure 10 for the echo filter. In both figures, different classes of the signals are represented by different colors: black: esa; green: fao; yellow: fax; red: fbo; blue: fbx. In Figure 9C, some yellow pixels (fax) can be seen at the upper right end of the originally predicted signals. Since they are outside of the boundary for fax, they were identified as incorrect classification and removed by the filter.

In Figure 10, the secondary echoes can be clearly seen in the original ionogram and were originally predicted as the true signals (i.e., primary echoes) by the model (Panel C). Most of these echo pixels were successfully identified and removed by the filter, as can be seen in Panels D and E.

The scatter plot of the IoU vs. SNR for the prediction results is shown in Figure 11. It can be seen that a low IoU (<0.5) occurred predominantly for a smaller SNR (<1.5), while a high IoU occurred when SNR > 1.4, which is very close to the median SNR = 1.43. This shows that the noise had a serious impact on the model accuracy and that the model operation was reasonable.

It should be noted that the accuracy of the ionogram recovery might be increased by the application of deeper learning models with a larger number of layers and training parameters. This will be a subject of the next paper, where various models will be compared. The purpose of the present paper was to identify the reasons that can affect the performance of the signal recovery using the FC-DenseNet24 as a test model. We found the following main reasons:1.*The percentage of signal samples in the dataset:*A comparison of Table 3 and Table 4 revealed that the sample percentages for eo and ex were both lower than 10%, the percentage for fax was less than 40%, and that for fao was barely 40%. This suggests that the sample percentage in a dataset needs to be higher than 40% in order for our model to adequately learn the characteristics of the signal and be able to recover it from the noisy ionograms;2.*Mixing with other signals:*In addition to the sample percentage, another factor that can affect the model performance is whether the location of a signal is overlapping with other signal classes. Figure 9B,C show a clear example of an overlapping signal in our ground truth and predictions, respectively. Although the sample percentage of fbx, which was nearly 90%, was significantly higher than that of esa, which was slightly over 50%, esa was located distinctly away from any F signal classes. In contrast, fbo and fbx signals were often overlapped, which caused difficulty in distinguishing and separating them even by a human. In addition, F1 layers in the ionograms were often seen connected (continued) to F2 layers without a distinctive dividing point, affecting the determination of their critical frequencies. Both factors contributed to the unimpressive IoU for fbx signals;3.*C/A ratio:*As explained earlier, our “ground truth” occupied an area larger than the area of the true signal because we labeled around the outer edge of the signal. Therefore, even if the model perfectly predicted the true signal, the IoU, which compares the prediction with the “ground truth”, would still be less than one. The derivation in the earlier section showed that the lower the C/A, the lower the IoU of the perfect prediction. As can be seen from the ionogram layers in Figure 1, F1 layers usually had a lower C/A than Es layers, which can contribute to the lower IoU for fao and fax. Figure 12 shows the scatter plots for compact and elongated signals for Es layers and F2 layers, respectively. The elongated F2 signals show that they had a high IoU in the regions where C/A was high. For the Es signals, we see a similar result wherein the high IoU cases were associated with the high C/A cases. It is evident that there was an inevitable decrease in the IoU as a result of labeling for both the compact and the elongated cases of the signal shape.

Lastly, we discuss and compare our model to previous work performed by the Peru team. The Peru auto-encoder CNN has 12 layers and a total of 4385 parameters. FC-DenseNet24 is deeper, having 24 layers, and thus, has more parameters (690,886). The Peru auto-encoder was trained using 816 labeled ionograms, and our FC-DenseNet24 was trained on 6131 labeled ionograms. The Peru auto-encoder only outputs a binary image containing the segmented signal from an ionogram that is unknown to the model. FC-DenseNet 24 on the other hand outputs a vector of 11 elements corresponding to different kinds of ionospheric signals. This makes our model capable of classifying the ionospheric layers. This more sophisticated scheme of our model was due to the large statistics of the Taiwan dataset. The design of FC-DenseNet24 is much more powerful and provides an improved accuracy in comparison to the auto-encoder. It is also important to note that the two models were trained using two different datasets: the Peru auto-encoder was trained on Peru ionograms; FC-DenseNet24 was trained on Taiwan ionograms. A more rigorous comparison between the two models would require training both of them using the two datasets and comparing the results. Nevertheless, FC-DenseNet was still able to provide an improved accuracy compared to the Peru auto-encoder despite having been trained on a much higher noise in the Taiwan dataset.

## 5. Conclusions

In this work, we developed an artificial neural network, FC-DenseNet24, based on the FC-DenseNet103 CNN [15], and applied it to automatically recover useful signals from Taiwan ionograms. For this purpose, we labeled 6131 ionograms manually and distinguished 11 different classes of the signals according to their ionospheric layers. We also developed a double-filtering algorithm to improve the accuracy of recovery by reducing incorrectly classified signal layers and the secondary echoes identified as true signals. The results showed that FC-DenseNet24 with filtering can successfully recover sporadic E layers and the F2 layers’ ordinary mode from highly noise-contaminated Taiwan ionograms (mean SNR = 1.43). The IoU of the recovery of these two signal classes was greater than 0.6, which is higher than the previous models reported.

We identified three factors that can lower the recovery accuracy: (1) the signal classes with insufficient samples in the dataset; (2) the signals that are mixing and overlapping with other signals in the ionogram; (3) the signals with a more compact shape, that is smaller Circumference-over-Area (C/A) ratio. Our results indicated that the IoU of the model predictions tends to be higher for signals that have a high C/A ratio.

## Figures and Tables

**Figure 1 sensors-21-06482-f001:**
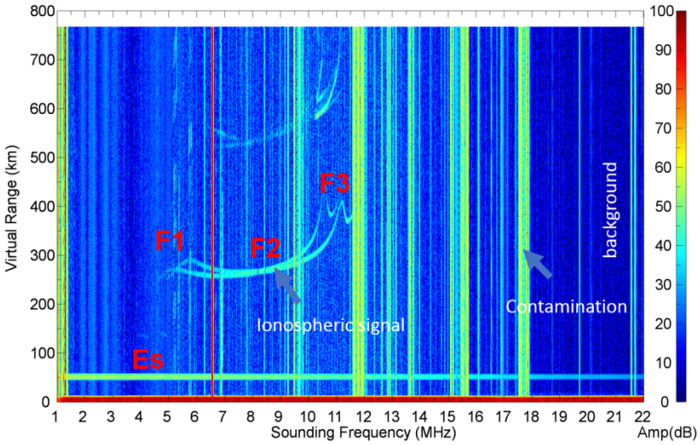
An ionogram containing three key features: the background, the contamination, and the ionospheric signal. The color bar on the right represents the amplitude of the signal in Decibels (dB).

**Figure 2 sensors-21-06482-f002:**
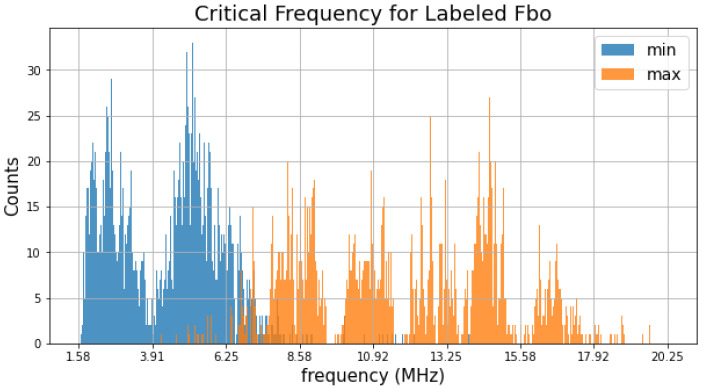
Distributions of the maximum (blue) and minimum (orange) frequencies of the labeled F2 layer of the ionograms.

**Figure 3 sensors-21-06482-f003:**
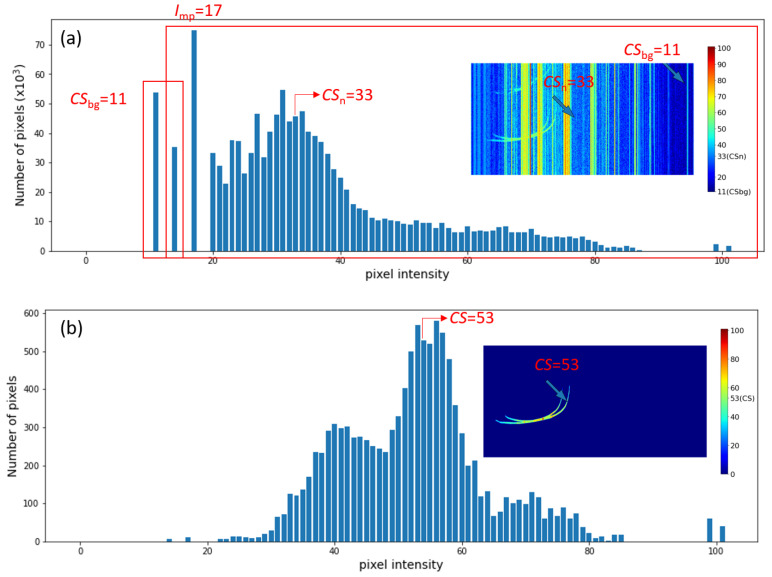
(**a**) The histogram of the pixel intensity *I* of the entire ionogram inserted on the right. (**b**) The histogram of the pixel intensity of the useful signal of the ionogram inserted on the right.

**Figure 4 sensors-21-06482-f004:**
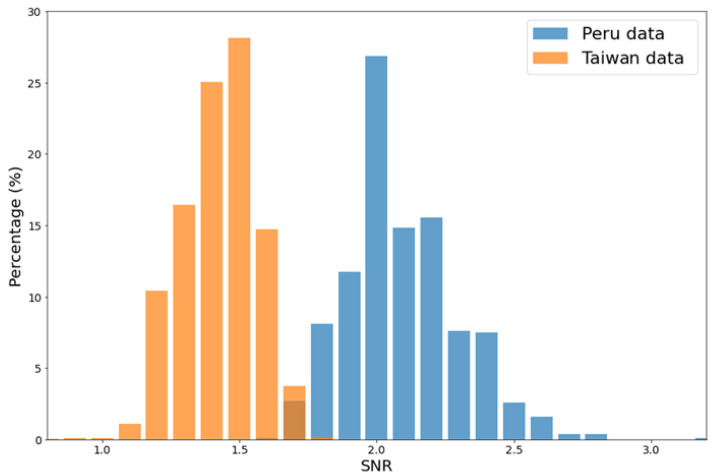
The histograms of the SNR for Peru data (blue) and for Taiwan data (orange). The mean SNR for Peru data is 2.09, and that for Taiwan data is 1.43, as indicated in the plot.

**Figure 5 sensors-21-06482-f005:**
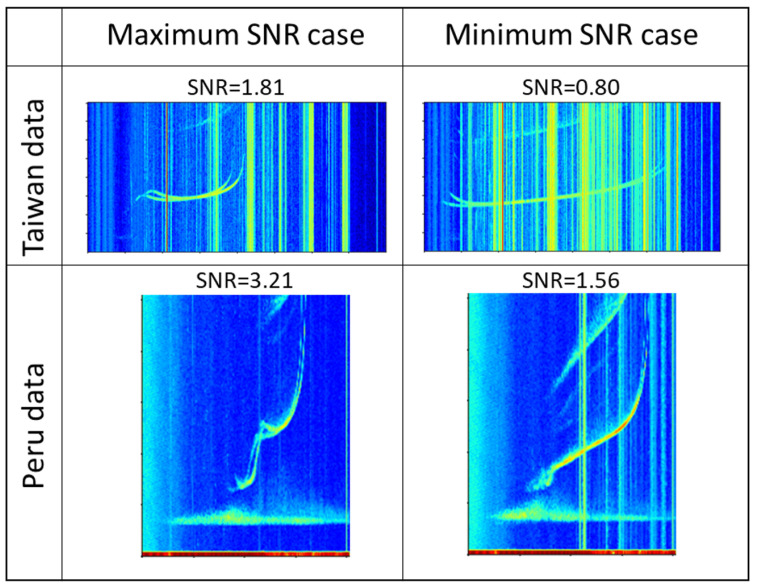
Maximum and minimum SNR cases for Taiwan (**upper panels**) and Peru (**lower panels**).

**Figure 6 sensors-21-06482-f006:**
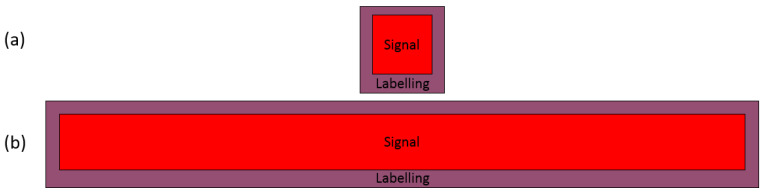
The two types of ionogram signals: (**a**) compact and (**b**) elongated. The area of the signal is shown in red, and the labeled area is shown in purple.

**Figure 7 sensors-21-06482-f007:**
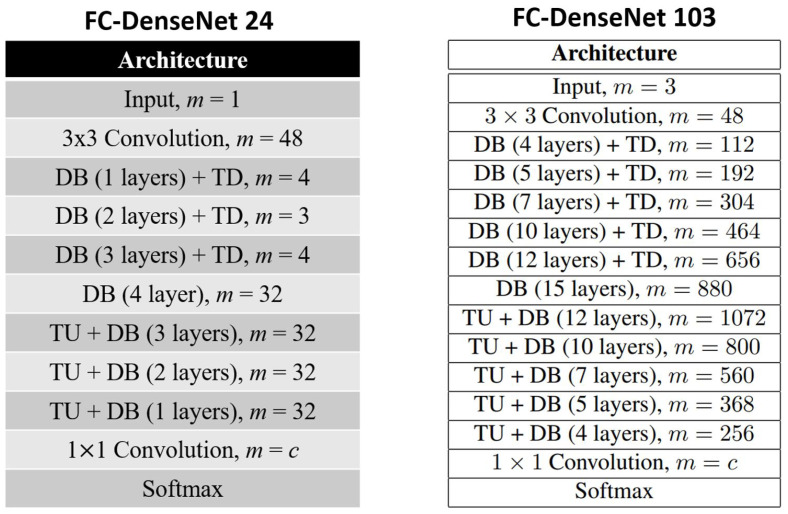
The architectures of the FC-Dense24 and FC-DenseNet103 model.

**Figure 8 sensors-21-06482-f008:**
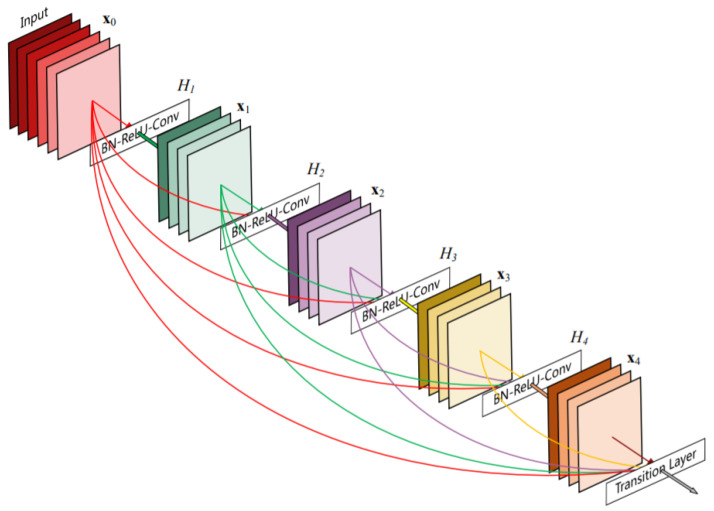
An illustration of a dense block.

**Figure 9 sensors-21-06482-f009:**
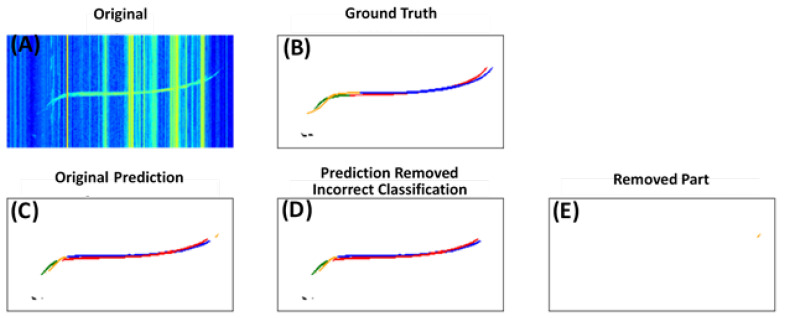
An example of filtering out incorrectly classified pixels. Different colors in (**B**–**E**) represent different classes of the signals: black: esa; green: fao; yellow: fax; red: fbo; blue: fbx. (**A**) original input ionogram; (**B**) labeled ground truth; (**C**) model prediction without filtering; (**D**) output after the filtering; (**E**) the incorrectly classified fax pixels removed by the filter.

**Figure 10 sensors-21-06482-f010:**
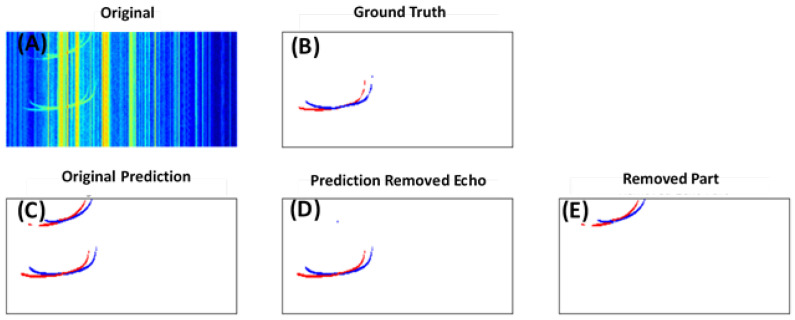
An example of filtering out secondary echoes. Different colors in (**B**–**E**) represent different classes of the signals: black: esa; green: fao; yellow: fax; red: fbo; blue: fbx. (**A**) Original input ionogram; (**B**) labeled ground truth; (**C**) model prediction without filtering; (**D**) output after the filtering; (**E**) the secondary echoes removed by the filter.

**Figure 11 sensors-21-06482-f011:**
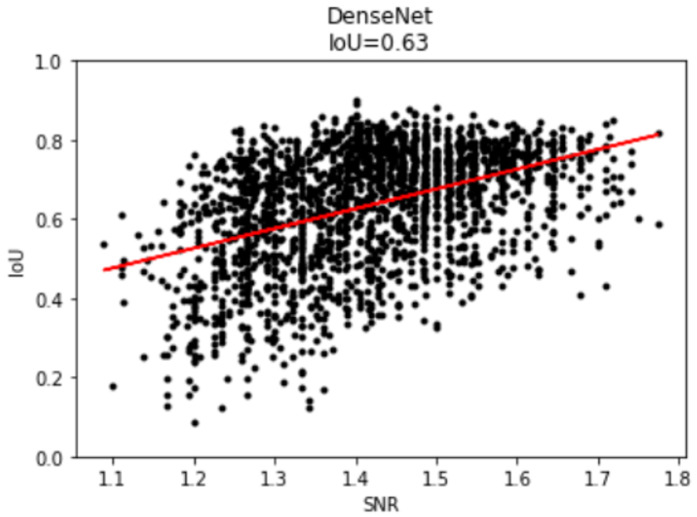
The scatter plot of IoU vs. SNR for all the predictions.

**Figure 12 sensors-21-06482-f012:**
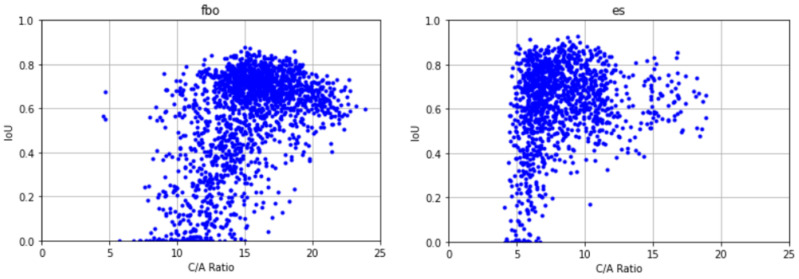
The scatter plots of IoU vs. the C/A ratio for the F2 and Es layers.

**Table 1 sensors-21-06482-t001:** The 11 ionospheric layers with their corresponding class labels.

Ionospheric Layer	Class Label
E layer ordinary mode	eo
E layer extraordinary mode	ex
Sporadic E layer ordinary mode	eso
Sporadic E layer extraordinary mode	esx
Sporadic E layer	es
F1 layer ordinary mode	fao
F1 layer extraordinary mode	fax
F2 layer ordinary mode	fbo
F2 layer extraordinary mode	fbx
F3 layer ordinary mode	fco
F3 layer extraordinary mode	fcx

**Table 2 sensors-21-06482-t002:** The composition layers of the basic building blocks of FC-DenseNet24. From left to right: Dense Block (DB), Transition Down (TD), and Transition Up (TU).

Dense Block (DB)	Transition Down (TD)	Transition Up (TU)
Batch Normalization	Batch Normalization	3 × 3 Transposed Convolution (*stride* = 2)
ReLU	ReLU	
3 × 3 Convolution	1 × 1 Convolution	
Dropout (*p* = 2)	Dropout (*p* = 2)	
	2 × 2 Max Pooling	

**Table 3 sensors-21-06482-t003:** The percentage of ionograms that contain the indicated signal class in the training, testing, and validation sets.

	eo	ex	eso	esx	es	esa	fao	fax	fbo	fbx	fco	fcx
Train	7.6	1.1	12.6	10.0	39.9	51.9	39.3	26.4	94.6	87.8	0.15	0.13
Test	9.2	1.1	14.8	11.2	40.8	55.1	38.7	25.5	94.2	87.7	0.08	0.08
Validation	7.5	0.5	14.5	12.2	38.0	52.1	39.4	25.3	95.5	89.8	0.10	0.40

**Table 4 sensors-21-06482-t004:** Comparison of the IoU between the original and the filtered outputs.

	eo	ex	eso	esx	es	esa	fao	fax	fbo	fbx	fco	fcx
Original	0.00	0.00	0.37	0.16	0.51	0.56	0.47	0.33	0.59	0.48	0.00	0.00
Filtered	0.00	0.00	0.38	0.16	0.57	0.66	0.47	0.33	0.61	0.50	0.00	0.00

## Data Availability

The data and models presented in this study are openly available on Kaggle.com. The links are provided as follows. For the data: https://www.kaggle.com/changyuchi/ncu-ai-group-data-set-fcdensenet24 (Ionogram Data Accessed on 5 August 2021); for the filtering: https://www.kaggle.com/changyuchi/filter (Filtering Accessed on 5 August 2021); for FC-DenseNet24: https://www.kaggle.com/changyuchi/fc-densenet24 (FC-DenseNet24 Accessed on 5 August 2021).

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
