# Peer review of "Recovery of Ionospheric Signals Using Fully Convolutional DenseNet and Its Challenges"

_sensors, 2021, doi:10.3390/s21196482_

Round 1

Reviewer 1 Report

The paper uses the Dense Net-based DL model for Ionospheric Signals recovery, which is really interesting. The authors also collect the datasets and perform evaluations. The quality of this paper could be considered for publication if the authors address the following issues:

  • The background of other pre-trained deep learning and their application areas could be presented in the paper. This would highlight the strengths of DL models in other domains as well and we could claim its efficacy in this domain as well based on it.  So, the authors could add some papers in their paper such as:
  •  1. Sitaula, Chiranjibi and Hossain, Mohammad Belayet 2020, Attention-based VGG-16 model for COVID-19 chest X-ray image classification, Applied Intelligence, pp. 1-14, doi: 10.1007/s10489-020-02055-x.
  • 2. Boulch, Alexandre, Noëlie Cherrier, and Thibaut Castaings. "Ionospheric activity prediction using convolutional recurrent neural networks." arXiv preprint arXiv:1810.13273(2018).
  • 3. Ulukavak, Mustafa. "Deep learning for ionospheric TEC forecasting at mid-latitude stations in Turkey." Acta Geophysica2 (2021): 589-606.
  • The authors need to highlight the reasons of using DenseNet instead of other DL models such as VGG-16, ResNet, EfficientNet, etc. in the current study.
  • The authors could compare the efficacy of current work using various pre-trained DL models
  • The comparison of present work with state-of-the-art may be carried out.
  • The motivations, for example open questions, in the introduction could be made clear.

Author Response

Reply to the Comments of Reviewer 1

We are very grateful to the Reviewer for very useful comments and recommendations. We have revised the manuscript in order to address most of them. In the revised manuscript, our corrections are marked by the red color.

Comment 1) The background of other pre-trained deep learning and their application areas could be

presented in the paper. This would highlight the strengths of DL models in other domains

as well and we could claim its efficacy in this domain as well based on it. So, the authors

could add some papers in their paper such as:

  1. a) Sitaula, Chiranjibi and Hossain, Mohammad Belayet 2020, Attention-based VGG-16 model for

COVID-19 chest X-ray image classification, Applied Intelligence, pp. 1-14, doi:

10.1007/s10489-020-02055-x.

  1. b) Boulch, Alexandre, Noлlie Cherrier, and Thibaut Castaings. "Ionospheric activity prediction using convolutional recurrent neural networks." arXiv preprint arXiv:1810.13273 (2018).
  2. c) Ulukavak, Mustafa. "Deep learning for ionospheric TEC forecasting at mid-latitude stations in Turkey." Acta Geophysica 69.2 (2021): 589-606.

Reply: We appreciate the Reviewer for the recommendation of very useful references. We discuss these papers in Introduction:

Lines 59-69: “In recent years, deep learning techniques have been implemented to aid ionospheric research. Predictive models based on deep learning methods are developed in order to forecast GPS-TEC at mid-latitudes in Turkey \cite{Mustafa2013}. In another similar study, convolutional neural networks (CNNs) have been also used to forecast TEC values globally \cite{Alexandre2013}. Deep learning has also played an important role in the recent Coronavirus Disease 2019 (COVID-19) pandemic. In particular, the attention-based VGG-16 model has been used as an image classification tool for COVID-19 diagnosis \cite{Sitaula2020}.  Recently, CNNs have been used to automatically recover the signals of the ionospheric F-layer from experimental ionogram data in Peru \cite{DLJS2019}. The results in that paper  showed that CNN models significantly removed the noise from the recovered ionograms and thus the quality of information was improved.”

Comment 2) The authors need to highlight the reasons of using DenseNet instead of other DL models such as VGG-16, ResNet, EfficientNet, etc. in the current study.

Reply: The present paper is mainly devoted to formulation of the key problems in the application of DL for the recovering of ionograms in Taiwan. In order to stress this key issue we have slightly modofoed the Title: “Recovery of Ionospheric Signals Using Fully Convolutional DenseNet and its Challenges”. In addition, we have revised the end of Introduction:

Lines 70-84: “In this study, we approach the problem of ionogram recovery by implementing a CNN model known as FC-DenseNet \cite{Jegou2017}. We show numerically that ionogram data obtained from Taiwan ionosondes have far more noise and contaminations as compared to Peru data. In addition, this image segmentation technique, which separates the wanted signals such as the ionogram signals from the unwanted signals such as the background and the noise, can also provide a classification between signals from various ionospheric layers as different classes. We demonstrate that the model has different capabilities for prediction of different classes because of various reasons, which are identified and discussed.

The goal in this work is to present the application of an advanced CNN model for improvement of the ionogram recovery techniques as well as to present several challenges upon applying the model for the complex ionogram data. Section 2 represents the experimental data and its main characteristics and problem features. The CNN model is introduced in Section 3. Section 4 describes the results of model application and discusses the problems for future work. Section 5 is conclusion.”

The comparison of various models will be a subject of the next paper, which we are preparing right now in order to submit soon to the Sensor Journal. We mentioned this in the Revised manuscript:

Lines 311-316: “It should be noted that the accuracy of the ionogram recovery might be increased by application of deeper learning models with larger number of layers and training parameters. This will be a subject of the next paper, where various models will be compared. The purpose of the present paper is to identify the reasons that can affect the performance of the signal recovery using the FC-DenseNet24 as a test model.  We have found the following main reasons:”

Comment 3) The authors could compare the efficacy of current work using various pre-trained DL models

Reply: We have revised the Section 4. Results. In Table 4, we present IoU for 11 classes. We also discuss the comparison in the revised text:

Lines 284-295: “The Peru group \cite{DLJS2019} applied auto-encoder to recover the signals from the ionospheric F-layer without distinguishing different classes (such as F1, F2 and F3), and yield IoU $= 0.57$. In contrast, our model can predict E and Es layers in addition to the F1, F2, and F3 layers. The former ones have very different geometry and location from those for the F layer. Because of that, a simple binary classification cannot be applied for the comparison of the two models. Instead, we have calculated the prediction of our model for the whole F layer that is the combination of the following classes: F = fao + fax + fbo + fbx + fco + fcx. As a result, we have got IoU $= 0.7$. Apparently, this accuracy is much higher than that for the Peru auto-encoder model. Our results show that despite operating on the ionograms with much lower SNR and needing to distinguish different  classes of signals, FC-DenseNet24 model is capable of producing a much higher level of accuracy for the F layer than that of Peru's auto-encoder.”

Comment 4) The comparison of present work with state-of-the-art may be carried out.

Reply: At the present time, only the Peru model is known for the recovery of ionograms (see above). As we already noticed, the comparison of various models will be a subject of the next paper.

Comment 5) The motivations, for example open questions, in the introduction could be made clear.

Reply: In the revised Introduction, we have clarified the key points and questions of our study (see above).

Reviewer 2 Report

The article discussed the identification and classification of signals from different ionospheric layers through filtering and deep learning model in a data set of 6131 manually labeled ionograms acquired from low-latitude ionosondes in Taiwan. I have the following comments

  1. The work is interesting and of practical importance.
  2. The authors have modified the FC-DenseNet103 to FC-DenseNet24 and said that “In order to further improve the training efficiency of this model, we modified some layers, structures, and parameters”. However, I cannot see the details of those modifications and the technical ground for those modifications. The authors should discuss the modification and the technical reason for each modification in the revised manuscript.
  3. In the reasons for low performance, the percentage of 30 to 40% is fairly high in classification problems. The low classification accuracy may be due to the adopted modified version of the FC-DensNet103, the authors should comment on the possibility of other classifiers for good results.
  4. Although, the 11 classes problem is more complex compared with the binary class problem of the Peru auto-encoder. A more effective way of showing the comparison would be to show the binary classification performance of the FC-DenseNet-24 proposed in the current work with the binary classification performance of the Peru auto-encoder.
  5. The comparison with the Peru auto-encoder should be using the same metrics of evaluation, not just in wording.
  6. I see the classification results for only 7 classes, the authors should add more details on the prediction performance of the 11 classes, their names, their correctly classified percentage, etc.

Author Response

Reply to the Comments of Reviewer 2:

We very appreciate the Reviewer for useful comments. In the revised version of the manuscript, we have done our best to address them. In the revised manuscript, our corrections are marked by the red color.

Comment 1.    The authors have modified the FC-DenseNet103 to FC-DenseNet24 and said that "In order to further improve the training efficiency of this model, we modified some layers, structures, and parameters". However, I cannot see the details of those modifications and the technical ground for those modifications. The authors should discuss the modification and the technical reason for each modification in the revised manuscript.

Reply: We have revised this part:

  1. Figure 7 now compare the structures of the FC-Dense24 and FC-DenseNet103 models.
  2. We add the following explanation in the text:

Lines 223-234: “Since one of the purpose of our work is to make the results be open for everybody, we have provided our model and data set to Kaggle (https://www.kaggle.com/), a web-based data and computing environment. The original version of the FC-DenseNet-103 model (see Figure ~\ref{fig:DenseNet2}, right panel) was too heavy to be run at Kaggle because of computing and memory limitations. Note that the resolution of Taiwan ionograms is 1600x800 pixels and the number of samples is more than 6000. These data require a lot of space, hence, we simplified the original model step-by-step in order to fit the requirements in Kaggle. As a result, we have achieved a configuration of the FC-DenseNet-24 model (see Figure ~\ref{fig:DenseNet2}, left panel), which is friendly for Kaggle such that everybody can run the model, verify our results and check for other data sets.  For the adjustments, we have reduced the number of blocks by two times and the number of layers by several times such that the number of training parameters was reduced from ~9.4 million to 690,886.”

Comment 2.   In the reasons for low performance, the percentage of 30 to 40% is fairly high in classification problems. The low classification accuracy may be due to the adopted modified version of the FC-DensNet103, the authors should comment on the possibility of other classifiers for good results.

Reply: We have to clarify that the 40% is not percentage of classification (a ratio of correct prediction to total number). This is Intersection over Union (IoU), which includes both qualitative classification and quantitative identification of the pixels belonging to different signal. In the revised manuscript, we modify the Tables with IoU in order to avoid the misunderstanding. As one can see in Table 4, the IoU for fbo signal is 0.59 that is higher than the Peru’s IoU = 0.57. Hence the deeper learning provides higher accuracy even for more noisy data. In order to stress this result we have calculated IoU for the whole F-layer in order to compare it with the Peru model (see below).

We also discuss the possibility of other classifiers for good results as the following:

Lines 311-316: “It should be noted that the accuracy of the ionogram recovery might be increased by application of deeper learning models with larger number of layers and training parameters. This will be a subject of the next paper, where various models will be compared. The purpose of the present paper is identification of the reasons that can affect the performance of the signal recovery using the FC-DensNet24 as a test model. We have found the following main reasons:”

Comment 3.    Although, the 11 classes problem is more complex compared with the binary class problem of the Peru auto-encoder. A more effective way of showing the comparison would be to show the binary classification performance of the FC-DenseNet-24 proposed in the current work with the binary classification performance of the Peru auto-encoder.

Comment 4.   The comparison with the Peru auto-encoder should be using the same metrics of evaluation, not just in wording.

Reply: The Peru model was constructed for prediction of the ionospheric F layer. Our model can predict E and Es layers in addition to the F layer. The former ones have very different geometry and location from those for the F layer. Because of that, a simple binary classification cannot be applied for the comparison of the two models. Instead, we have calculated the prediction of our model for the whole F layer that is the combination of the following classes: F = fao + fax + fbo + fbx + fco + fcx. As a result, we have got IoU = 0.7. Apparently, this accuracy is much higher than that for the Peru auto-encoder model (IoU=0.57). This fact indicates clearly the advantage of a deep learning model.

In the revised manuscript, we write the following text:

Lines 280-296: “

In Table~\ref{tab4}, we see that the model is capable of identifying the es and fbo signals with an accuracy of IoU $> 0.5$. By applying the double-filtering algorithm, the accuracy is further increased to IoU $> 0.6$.  The recovery of eso, esx, fao, fax, and fbx signals are between $0.2 < $ IoU $ < 0.5$. The eo, ex, fco, and fcx classes have an IoU of 0.

The Peru group \cite{DLJS2019} applied auto-encoder to recover the signals from the ionospheric F-layer without distinguishing different classes (such as F1, F2 and F3), and yield IoU $= 0.57$. In contrast, our model can predict E and Es layers in addition to the F1, F2, and F3 layers. The former ones have very different geometry and location from those for the F layer. Because of that, a simple binary classification cannot be applied for the comparison of the two models. Instead, we have calculated the prediction of our model for the whole F layer that is the combination of the following classes: F = fao + fax + fbo + fbx + fco + fcx. As a result, we have got IoU $= 0.7$. Apparently, this accuracy is much higher than that for the Peru auto-encoder model. Our results show that despite operating on the ionograms with much lower SNR and needing to distinguish different classes of signals, FC-DenseNet24 model is capable of producing a much higher level of accuracy for the F layer than that of Peru's auto-encoder.”

Comment 5.    I see the classification results for only 7 classes, the authors should add more details on the prediction performance of the 11 classes, their names, their correctly classified percentage, etc.

Reply: We have done a major revision of this part of Results. In Tables 3 and 4, we show the results for 11 classes. On the base of those results, we explain why we combine eso, esx snd es labels into one esa label and why we exclude the F3 layer (fco & fcx) from the consideration:

Lines 274-276: “Because of low percentages of the eso and esx classes, the three classes assigned to the Es layer (eso, esx, es) are combined into one class "esa", as presented in Tables 3 and 4.”

Round 2

Reviewer 1 Report

Given the authors' effort to address the reviewer's concern, the reviewer is included to accept the current version.

Reviewer 2 Report

Thank you for addressing my comments. I recommend the article for publication in Sensors Journal.